# Usefulness of ß-d-Glucan Assay for the First-Line Diagnosis of *Pneumocystis* Pneumonia and for Discriminating between *Pneumocystis* Colonization and *Pneumocystis* Pneumonia

**DOI:** 10.3390/jof8070663

**Published:** 2022-06-24

**Authors:** Jeanne Bigot, Sandra Vellaissamy, Yaye Senghor, Christophe Hennequin, Juliette Guitard

**Affiliations:** 1Centre de Recherche Saint-Antoine, CRSA, AP-HP, Inserm, Hôpital Saint-Antoine, Service de Parasitologie-Mycologie, Sorbonne Université, F-75012 Paris, France; jeanne.bigot@aphp.fr (J.B.); christophe.hennequin-sat@aphp.fr (C.H.); 2Hôpital Saint-Antoine, Service de Parasitologie-Mycologie, Sorbonne Université, F-75012 Paris, France; sandra.vellaissamy@aphp.fr (S.V.); yaye.senghor@aphp.fr (Y.S.)

**Keywords:** *pneumocystis* pneumonia, *pneumocystis* colonization, molecular diagnostic, BDG, HIV patients, non-HIV immunocompromised patients

## Abstract

According to the immunodepression status, the diagnosis of *Pneumocystis jirovecii* pneumonia (PjP) may be difficult. Molecular methods appear very sensitive, but they lack specificity because Pj DNA can be detected in Pneumocystis-colonized patients. The aim of this study was to evaluate the value of a serum ß-d-Glucan (BDG) assay for the diagnosis of PjP in a large cohort of HIV-negative and HIV-positive patients, either as a first-line diagnostic test for PjP or as a tool to distinguish between colonization and PjP in cases of low fungal load. Data of Pj qPCR performed on bronchopulmonary specimens over a 3-year period were retrieved retrospectively. For each result, we searched for a BDG serum assay performed within ±5 days. Among the 69 episodes that occurred in HIV-positive patients and the 609 episodes that occurred in immunocompromised HIV-negative patients, we find an equivalent sensitivity of BDG assays compared with molecular methods to diagnose probable/proven PjP, in a first-line strategy. Furthermore, BDG assay can be used confidently to distinguish between infected and colonized patients using a 80 pg/mL cut-off. Finally, it is necessary to search for causes of false positivity to increase BDG assay performance. BDG assay represents a valuable adjunctive tool to distinguish between colonization and infection.

## 1. Introduction

*Pneumocystis jirovecii* (Pj) is an opportunistic fungal pathogen responsible for life-threatening pneumonia [1]. Immunocompromised patients with uncontrolled HIV infection, solid organ transplantation, suffering hematological malignancies, or receiving high doses of corticosteroids are at high risk for Pj pneumonia (PjP) [2,3,4]. According to the underlying disease, the clinical and radiological presentation may be more or less evocative of PjP. Indeed, whereas most of HIV-positive patients present with a classic triad of symptoms (dyspnea, fever, dry cough), and chest or CT-scan showing bilateral ground-glass opacities, the picture is much more complex and polymorphic in HIV-negative patients [5]. Moreover, in those patients, hypoxemia and respiratory failure are more frequent, leading to worse prognosis as compared to that of HIV-positive patients [2,3,4,5].

Performance of diagnostic tools also varies according to the clinical context. This is due to the lower pulmonary fungal load observed in HIV-negative patients so that the demonstration of the pathogen (asci and/or trophic forms) after staining broncho-alveolar fluid or induced sputum specimens is much rarer in those patients [6,7]. Finally, the detection of Pj DNA, thanks to PCR-based methods applied to these samples, appears the most reliable approach for the diagnosis of PjP in HIV-negative patients. However, although they are very sensitive, molecular techniques are more difficult to interpret, even with the advent of quantitative PCR. Indeed, many studies have shown that some patients can be considered colonized with Pj, a condition where Pj DNA is detected in bronchopulmonary samples, despite patients not presenting with clinical or radiological signs of PjP [8,9,10,11,12]. To circumvent this problem, some authors have suggested the use of cut-offs to distinguish between colonization and true infection [6,13,14,15]. However, in addition to the fact that these values depend on the qPCR method used, it has been shown that HIV-negative patients may harbor a wide range fungal load, so that 18% to 28% of patients have a fungal load within a grey zone making the final diagnosis uncertain [6,13].

Thus, any new marker for the diagnosis of PjP, notably in HIV-negative patients, would be helpful. Β-d-Glucan (BDG) is a polysaccharidic component of the cell wall of many fungal species, including Pj [16]. Several studies, mostly conducted in HIV-positive patients, have shown that PjP is associated with a high level of serum BDG [17,18,19], suggesting that a higher cut-off of 100 µg/mL [20], 200 µg/mL [18,21] or 400 µg/mL [22] may improve BDG performance. Similar research, conducted in HIV-negative patients, described suboptimal sensitivity and specificity of serum BDG to diagnose PjP [23], with some studies also describing high BDG values in patients presenting PjP [24,25] and another describing low BDG values especially in hematological patients presenting PjP [26]. Furthermore, a strong correlation between BALF fungal load and serum BDG has been found in HIV-positive patients but was found to be weak in immunocompromised non-HIV patients especially in hematological patients [27,28].

In this study, we aimed to assess the diagnostic value of BDG for the diagnosis of PjP on a large cohort of immunocompromised patients, infected or not with HIV. In particular, we looked at the value of BDG assay as a first-line diagnostic test for PjP before Pj qPCR results are available, and its potential value for distinguishing between colonized and infected patients presenting with a low fungal load.

## 2. Materials and Methods

### 2.1. Patients

We first retrospectively selected all immunocompromised patients having been tested for Pj DNA in a broncho-alveolar lavage fluid (BALF) or an induced sputum (IS) specimen between 1 January 2018 and 31 December 2020 in a French University Hospital in Paris. Each sample was considered an episode if distant for more than 30 days from another one. We then selected episodes for which a serum BDG assay was performed between 5 days before and 5 days after the date of the bronchopulmonary sample.

### 2.2. Diagnostic Methods

Direct microscopy for the demonstration of Pj was only performed on BALF specimens. Cytospin smears were stained with Giemsa and CalcoFluor (Sigma Aldrich, St. Quentin Fallavier, France) and examined using light and fluorescent microscopy respectively, at an ×500 magnification.

The detection of Pj DNA was performed according to a previously published qPCR protocol targeting Pj mt LSU [6]. For HIV-positive patients, we considered a cut-off at 30,000 copies (cp) of the target/mL to distinguish between colonization and infection, whereas a fungal load <3000 cp/mL and >30,000 cp/mL was used for this purpose in HIV-negative patients. In between these values corresponded a grey zone with uncertain diagnosis [6]. For IS we used the same extraction qPCR methods as for BALF, and the same fungal load cut-off.

BDG was assayed using the Fungitell kit (Cape Code) according to the manufacturer’s recommendations. BDG values <60 pg/mL and >80 pg/mL were considered negative and positive respectively, and equivocal in between these values. No further dilutions were performed for BDG results ≥523 pg/mL. For each positive assay, we recorded the concomitance of another invasive fungal infection and potential cause(s) of false-positive results, including perfusion of blood-derived products (platelet, packed red blood cells, albumin, fresh-frozen plasma, polyvalent immunoglobulin), digestive lesions (severe mucositis, digestive graft versus host disease), use of cellulose-containing medical devices (hemodialysis, surgical compresses), and *Nocardia* sp. infection [29].

Proven PjP was defined based on the demonstration of *Pneumocystis* forms on direct microscopy of a BALF specimen. The diagnosis of probable PjP was applied to patients with a positive qPCR and several criteria including an underlying immunosuppressive condition, clinical symptoms and radiological signs deemed to be compatible with PjP by clinicians. Pj colonization corresponded to any case, not included in the previous two groups, which presented with a positive qPCR, whatever the fungal load. Finally, episodes with a negative qPCR and the lack of fungal form detected in BALF were considered as “non-PjP”.

### 2.3. Statistics

Comparisons of BDG values according to the clinical status were performed using Kruskal-Wallis multiple comparisons tests followed by Dunn’s pair-wise comparisons.

Correlations between qPCR and BDG values in the probable/proven PjP were performed using the Spearman correlation test. Youden index was calculated to evaluate the performance of the BDG assay. The Kappa coefficient was used to measure agreement between qualitative results of BDG value and qPCR.

All tests were performed using Prism version 6.0 (GraphPad), considering a two-tailed *p* value < 0.05 as significant.

### 2.4. Ethic

The study was performed retrospectively. All specimens were previously collected through routine clinical tests and patient-identifiable information was anonymized. According to the ethical standards of the French Ethics Committee on human experimentation and with the Helsinki Declaration of 1975 as revised in 2008, no written or verbal informed consent to participate in this study from patients was necessary.

## 3. Results

### 3.1. Patients and Clinical Samples

Between 1 January 2018 and 31 December 2020, Pj qPCR was performed on 3012 samples (2777 BALF and 235 IS) collected from 2316 immunocompromised patients. The result of a BDG assay performed ± 5 days around the date of the bronchopulmonary sample was available for analysis for 683 episodes (Figure 1). Clinical records were not found for 5 episodes that were therefore excluded from the analysis. Sixty-nine and 609 episodes occurred in 67 HIV-positive and 575 HIV-negative patients, respectively. Underlying conditions of patients at the time of episodes were mostly hematological malignancies (*n* = 229, 37.6%), immunosuppressive treatment (*n* = 217, 35.6%), solid organ transplantation (*n* = 90, 14.8%), cancer (*n* = 69, 11.3%), and congenital immunodeficiency (*n* = 4, 0.7%).

### 3.2. BDG Value for the Diagnosis of PjP in 69 Episodes from 67 HIV-Positive Patients

Direct microscopy was found positive for 17 episodes out of 69, corresponding to an incidence of proven PjP at 24.6%. In addition, 9 episodes (13%) were classified as probable PjP by clinicians, and, finally, 7 (10.1%) and 36 (52.2%) episodes were considered as Pj colonization and non-PjP cases, respectively. qPCR was found positive for 32 episodes (Appendix A). All but one proven or probable PjP had a positive qPCR. One patient presenting an episode with a proven PjP (demonstration of Pj forms in a BALF specimen) had a negative qPCR result which was found to be related to a point mutation in the probe target region (data not shown). There were two cases of discrepancy between the fungal load and the PjP categorization: one patient presenting an episode with a probable PjP had a positive qPCR but with fungal load below the retained cut-off (<30,000 cp/mL); while another one with an episode of Pj colonization (direct microscopy and clinical suspicion negative) had a fungal load >30,000 cp/mL (Appendix A). In the 65 episodes of proven or probable PjP with a positive qPCR, *Spearman* correlation between qPCR and BDG value retrieved an R = 0.11 with a non significant *p* value.

Thus, using the single previously defined cut-off at 30,000 cp/mL, sensitivity, specificity, positive predictive value (PPV) and negative predictive value (NPV) of the qPCR for the diagnosis of PjP (proven or probable) in HIV-positive patients were at 92.3% [95% CI: 75.9–98.6%], 97.7% [95% CI: 87.9–99.9%], 96% [95% CI: 80.5–99.8%] and 95.5% [95% CI: 84.9–99.2%], respectively, with a likelihood ratio of 13.2 (Table 1). Youden index and Kappa coefficient were calculated to be 0.9 and 0.59, respectively.

We first evaluated the BDG assay performances in this cohort of patients as a first-line test for the diagnosis of PjP episodes in patients with suspected PjP (*n* = 69). All 17 episodes of proven PjP had a positive BDG, including an episode with a negative qPCR (Figure 2a). All but one of the 9 episodes of probable PjP had a positive BDG result (>80 pg/mL). This patient was not receiving Pj treatment and/or Pj prophylaxis. Sensitivity to support the diagnosis of PjP in HIV-positive patients was calculated to be 96.1% [95% CI: 81.1–99.8%] using a BDG cut-off of 80 pg/mL (Table 1). Among the 43 episodes considered as non-PjP or Pj colonization, 8 (18.6%) presented with BDG > 80 pg/mL, leading to a specificity of 81.4% [95% CI: 67.4–90.3%]. Five out of these 8 patients had at least one evident cause of false positive BDG value at the time of the episode, such as invasive aspergillosis (*n* = 1), candidiasis (*n* = 1), polyvalent immunoglobulin infusion (*n* = 1) or PjP episode in the previous 2 months (*n* = 2). Excluding those episodes (Figure 2b) led to an increase of the specificity rate to 92.1% [95% CI: 79.2–97.3%] (Table 1).

The median value of serum BDG of patients with a Pj colonization or with non-PjP episodes (32.7 pg/mL in both cases) was significantly lower than that of patients with a proven or a probable infection (523 pg/mL in both cases) (Figure 2a). A BDG value > 80 pg/mL, >200 pg/mL and >400 pg/mL was found in 33, 28 and 24 episodes, respectively. Increasing the BDG cut-off at 200 pg/mL and 400pg/mL, and excluding episodes with evident causes of BDG false positive results, increased the specificity to 97.4 [95% CI: 86.8–99.9%] and 100% [95% CI: 91.4–100%], respectively; but with a sensitivity decreasing to 88.5% [95% CI: 71–96%] and 84.6% [95% CI: 66.5–93.8%], respectively (Table 1).

We then investigated the ability of the BDG assay to distinguish between colonized and Pj infected HIV-positive patients with a low fungal load as detected by qPCR. Seven episodes from 67 HIV-positive patients presented a fungal load <30,000 cp/mL, among which one was a probable PjP episode as considered by clinicians while the other six were considered to be Pj colonization. The patient presenting an episode of probable PjP had a BDG value > 523 pg/mL. Four episodes considered as Pj colonization were negative for serum BDG while the other two had positive serum BDG at 212 pg/mL and 414 pg/mL. Both patients had an evident cause for their “false-positive” result at the time of the episode (invasive aspergillosis *n* = 1, PjP episode in the previous 2 months (*n* = 1)). Sensitivity and specificity were not calculated considering the low number of episodes (*n* = 7).

### 3.3. BDG Value for the Diagnosis of 609 PjP Episodes in 575 HIV Negative Patients

A similar analysis was performed for HIV-negative immunocompromised patients. Among the 609 episodes from 575 patients, direct microscopy was found to be positive in 8 cases (proven PjP). Twenty-eight episodes were classified as probable PjP by clinicians, and, 46 and 527 episodes were considered as Pj colonization and non-PjP, respectively.

A Pj qPCR was found to be positive in 82 episodes (13.5%) (Appendix A). All episodes with a proven PjP (*n* = 8) and 27 among 28 episodes with a probable PjP had a positive qPCR with a median fungal load of 10^7^ cp/mL and 4.5 × 10^4^ cp/mL, respectively. One episode classified as probable PjP presented with a negative qPCR result, which was further found to be related to a point mutation in the probe target sequence. In addition, 13/36 (36%) episodes with a probable/proven PjP had a fungal load below the 30,000 cp/mL cut-off previously proposed. The median fungal load of colonized episodes (*n* = 46) was at 1419 cp/mL, a value significantly lower than that of proven and probable infection at 44,760 cp/mL and 10,208,400 cp/mL, respectively (Appendix A). In contrast, 15/46 (32.6%) episodes of Pj colonization presented with a fungal load higher than the 3000 cp/mL cut-off previously proposed [6]. To obtain the best sensitivity, the lower cut-off (3000cp/mL) should be used; in this case, sensitivity and NPV of the qPCR for the diagnosis of PjP episodes in HIV-negative patients were at 86.1% [95% CI: 71.3–93.9%] and 99.1% [95% CI: 97.9–99.6], respectively. Using the 30,000 cp/mL cut-off increased specificity and PPV to 99.7% [95% CI: 98.7–99.9%] and 92% [95% CI: 75–98.6%], respectively (Table 2). Youden index and Kappa coefficient were calculated to be 0.64 and 0.22, respectively.

In HIV-negative patients, a BDG value > 80 pg/mL, >200 pg/mL and >400 pg/mL was found in 132 (21.7%), 76 (12.5%) and 55 (9%) episodes, respectively. The BDG median value of 523 pg/mL for episodes occurring in patients with either a proven or a probable infection significantly exceeded the value observed in colonized or non-PjP episodes (median of 31 pg/mL for both) (Figure 3a). Among the 36 proven/probable PjP episodes, 32 (sensitivity at 88.9% [95% CI: 74.7–95.6%]) had a serum BDG value > 80 pg/mL, 23 (71.9%) of them having a serum value > 400 pg/mL, including an episode of probable infection with a negative qPCR (Figure 3). Reviewing clinical charts of the four (14.3%) episodes with a probable PjP and a negative BDG assay (<80 pg/mL), there was strong support for the diagnosis of PjP for three of them, while a PjP combined with an everolimus-induced pneumonia was the retained diagnosis for the fourth. In the 35 episodes of proven/probable PjP with a positive qPCR, Spearman correlation between qPCR and BDG value retrieved an R = 0.32 with a *p* value of *p* = 0.06.

Among the 573 episodes considered non-PjP or Pj colonization, 100 (17.5%), 50 (8.7%) and 32 (5.6%) episodes presented with a BDG value > 80 pg/mL, >200 pg/mL and >400 pg/mL, respectively, leading to a specificity ranging from 82.6% [95% CI: 79.2–85.4%] to 94.4% [95% CI: 92.2–96%] (Table 2). Concomitant fungal infection (*n* = 40), digestive breach (*n* = 11), dialysis (*n* = 9), fractioned-blood product (*n* = 13) and polyvalent immunoglobulin (*n* = 12) perfusion were found in those patients. According to the considered cut-off, this may have led to false positive results in 76, 42 and 29 episodes, respectively. Best specificity and PPV at 99.8% [95% CI: 99–100%] and 95.8% [95% CI: 79.8–99.8%], respectively, were obtained when excluding those episodes and using a 400 pg/mL cut-off (Table 2 and Figure 3b).

We then assessed the value of the BDG assay to distinguish colonization from PjP episodes, when qPCR was positive and <30,000 cp/mL (*n* = 56). Among these episodes, 12 and 44 episodes were considered as having probable PjP and Pj colonization, respectively. Of the 12 episodes with a probable PjP, 10 (83.3%), 7 (58.3%) and 5 (41.7%) had a BDG value > 80 pg/mL, >200 pg/mL and >400 pg/mL, respectively. Among the 44 episodes of colonization, there were 10 (22.7%), 4 (9%) and 2 (4.5%) episodes with a positive BDG level >80 pg/mL, >200 pg/mL and >400 pg/mL, respectively. An evident cause for a false-positive BDG result occurred in 6 of them: hemodialysis (*n* = 2), polyvalent immunoglobulin perfusion (*n* = 1), blood-derived products perfusion (*n* = 2), digestive perforation (*n* = 1), and invasive aspergillosis (*n* = 1). Using an 80 pg/mL cut-off, sensitivity and specificity of the BDG assay to distinguish between colonization and infection in HIV-negative patients were at 83.3% [95% CI: 55.2–97%] and 77.3% [95% CI: 63–87.2%], respectively. Specificity reached 100% [95% CI: 91.6–100%] when using a 400 pg/mL cut off and excluding from the analysis episodes with an evident false positive result, but sensitivity was reduced to 41.7% [95% CI: 19.3–68%] (Table 3). Of note, a patient with a probable PjP and a negative qPCR assay had a BDG value of 523 pg/mL.

## 4. Discussion

Several studies have pointed out the differences in the clinico-radiological presentation of PjP according to HIV-serological status [2,3,4,5,6,7]. Notably, HIV-negative patients frequently present with a more severe form that may preclude the realization of BAL and have a worse prognosis [30]. qPCR results obtained from HIV-negative are more widely spread, so that 36% of HIV-negative patients with a probable/proven PjP had a fungal load below a cut-off of 30,000 cp/mL, making a diagnosis unreliable when based only on the qPCR result. Altogether, these conditions make the diagnosis of PjP much more difficult in HIV-negative patients as compared to HIV-positive patients.

Several studies conducted in HIV-positive patients have demonstrated the usefulness of BDG for a non-invasive diagnosis of PjP [20,21]. In HIV-negative patients, the ECIL recommended the use of BDG to exclude PjP in a screening strategy because of a high NPV [31]. However, recent studies pointed out the non-optimal sensitivity of the test calculated at 87% in those patients, and even worse in onco-hematological patients with a 64% sensitivity [26]. Negative predictive value can be high but mostly reflects the low incidence in the populations of HIV-negative immunocompromised hosts.

In this study, we retrospectively evaluated the performance of BDG assays with a focus on two clinical indications: (i) as a first-line diagnostic test in immunocompromised patients suspected of PjP before the Pj qPCR result is available; and (ii) as an aid to distinguishing between Pj colonization and infection in patients presenting with a low fungal load detected with qPCR applied to a bronchopulmonary sample.

Our study highlights that BDG and qPCR sensitivity was quite similar as a first-line diagnostic test in HIV-positive or HIV-negative patients with presumed PjP (92.3 vs. 96.1%, and 86.1% vs. 88.9%, for episodes occurring in HIV-positive and HIV-negative patients, respectively) but with a specificity much lower than the qPCR (81.4% vs. 93%, and 82.6% vs. 97.4%, respectively). This is due in part to the fact that two of the episodes with proven and probable PjP were not detected by qPCR because of mutation in the molecular target. This has already been described but with a much lower prevalence [32]. However, these results are overall in accordance with a meta-analysis that calculated sensitivity and specificity of BDG to be 94% and 83%, respectively, for HIV-positive patients; and to be 86% and 83%, respectively, for HIV-negative patients [23]. In addition to the fact that BDG is a “pan-fungal” marker, so can be positive in the case of other concomitant invasive fungal infection, this assay is subject to a variety of interference that decreases the specificity of the test [29]. According to the HIV-serological status, we found that 7.3% of episodes for HIV positive, and 13.5% for HIV-negative, occurred in patients with at least one condition that may lead to a positive BDG assay, respectively. Indeed, HIV-negative patients are more prone to develop other fungal infections that can lead to a positive BDG result. In addition, the clinical context in these immunocompromised patients frequently leads to the prescription of blood fractioned products (intravenous immunoglobulins,…), intestinal toxic drugs that may also be a source of false positive BDG, or other surgical procedures [29]. Excluding those patients from the analysis increases the specificity of BDG assay to 92.1% and 96.3% for episodes occurring in HIV-positive and HIV-negative patients, respectively, which are values quite similar to the qPCR specificity at 93% and 97.4%, respectively.

There were two cases of discrepancy between the fungal load and PjP categorization in HIV-positive patients. The first occurred in an HIV-positive patient (184 CD4/µL) with typical ground glass opacities, who was considered PjP. Both serum BDG assay (>523 pg/mL) and BALF that allow the detection of a low-copy number of Pj DNA (1843 cp/mL) were performed distantly (14 days) from initiation of an atovaquone therapy that allowed rapid and total recovery. The second episode occurred in an HIV-positive patient (129 CD4/µL), presenting with an acute lobar pneumonia and a high fungal load (628,800 cp/mL). Serum BDG assay was positive (391 pg/mL). Nevertheless, the patient was considered colonized and treated with amoxicillin-clavulanic acid allowing a favorable outcome.

Differentiating PjP and Pj colonization is important, as sulfamethoxazole-trimethoprim treatment, the first-line therapy for PjP, is associated with a high number of severe side effects [33]. Whether colonized patients should be treated remains a subject of debate but lower posology (prophylactic dosage), which is better tolerated, can be proposed [34]. Considering episodes occurring in HIV-positive patients, only seven presented with Pj low fungal load, thus, results should be interpreted with caution. Nevertheless, BDG was able to correctly categorize episodes in five cases out of seven (71.4%); and in five out of five (100%) if the two episodes occurring in patients presenting evident causes of false positive BDG were excluded. For episodes occurring in HIV-negative patients with a low fungal burden in BALF (*n* = 56), the sensitivity of BDG assay was at 83.3%. Among these cases, we found six cases where false-positive BDG results could be suspected. The specificity increased from 77.3% (when including those episodes in the analysis) to 89.5% (when excluding them). Thus, our analysis supports the use of BDG to distinguish between Pj colonization and PjP in low fungal load patients, when excluding patients harboring conditions that may lead to a positive BDG assay. Very few studies have analyzed the value of BDG to distinguish between colonized and infected Pj patients with low fungal load. In a recent study, Liu et al. found that BDG assays correctly classified 25 patients out of 27 patients with low Pj fungal burden as either Pj colonized or PjP. In their study, quantification of Pj fungal burden relied on the quantification of Pj reads obtained thanks to a metagenomics approach applied to BAL [35].

Thus, as very recently published by Lagrou et al. [36], BDG has adjunctive value for distinguishing the probable PjP from the colonized patient, if causes of false positive results have been ruled out.

The retrospective nature of our study is a limitation in the interpretation of the results. Indeed, clinicians used the results of BDG to categorize the patients, which may have introduced a bias. Few pediatrics HIV-negative episodes (29 out of 609, 4.8%) have been included in the study. Moreover, we detect 1 episode of probable PjP and 28 episodes free of Pj, among which prevalence of false positive BDG assays was calculated at 17.85%, very similar to the prevalence in the whole HIV-negative cohort (17.5%). Our study points out the fact that each positive BDG assay requires a fine analysis of the clinical charts to exclude from the interpretation, cases with known causes that can result in false-positive results for BDG assay. Most often, this detailed analysis can only be done distantly from the date of BDG assay, notably for the exclusion of diagnosis of other invasive infections. Finally, a small proportion of patients, 36 among 642 (5.6%), and only in the non-PjP group, presented with two episodes of suspected Pj and this may have introduced a bias, probably limited to the diagnostic efficacy calculation.

## 5. Conclusions

In conclusion, whatever the HIV-serological status, a BDG assay may be used to anticipate the diagnosis in patients with presumed PjP before the result is available from a qPC performed on their bronchopulmonary sample. Furthermore, the use of BDG assay to distinguish between colonized and PjP in patients presenting low fungal load is useful. A complete mycological check-up and a complete review of health-care associated factors should be done to exclude potential causes of false positive BDG results.

## Figures and Tables

**Figure 1 jof-08-00663-f001:**
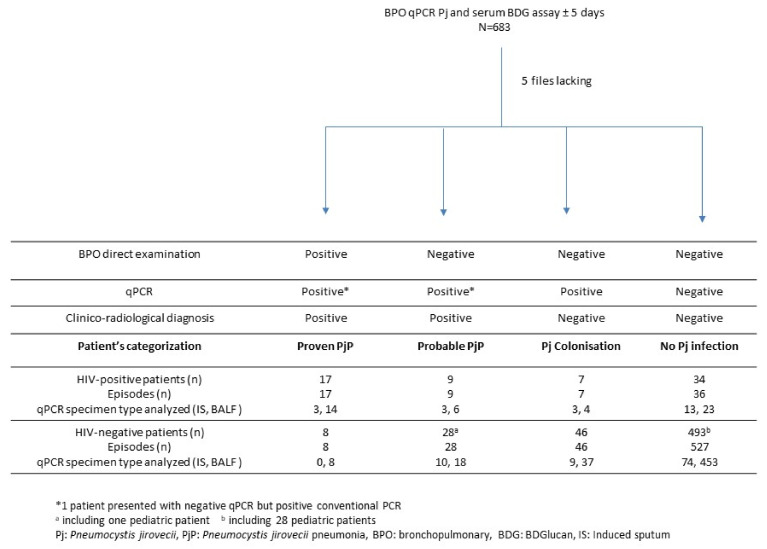
Characteristics and classification of patients/episodes included in the study.

**Figure 2 jof-08-00663-f002:**
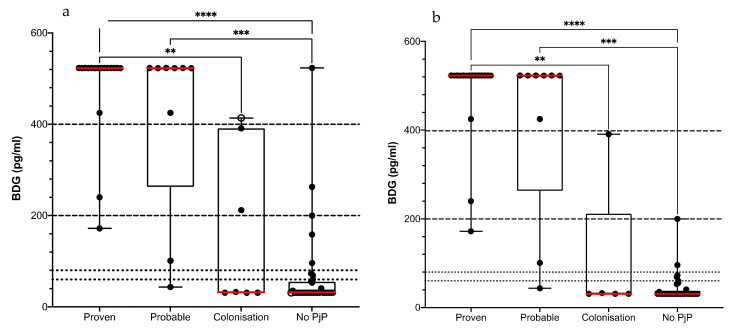
BDG value in 69 episodes from 67 HIV-positive patients suspected of PjP according to the final diagnosis retained. BDG values (**a**) with and (**b**) without evident cases of BDG false positive results. Comparison of the four groups was done using Kruskal-Wallis tests followed by Dunn’s pair-wise comparisons; *p*-adjusted values are indicated. Red line: median BDG value. ** *p* < 0.01, *** *p* < 0.001, **** *p* < 0.0001.

**Figure 3 jof-08-00663-f003:**
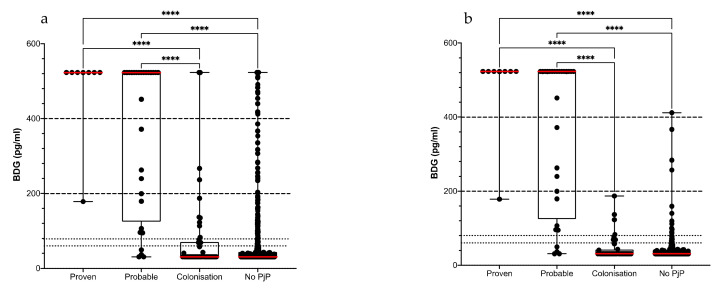
BDG value in 609 episodes from 575 HIV-negative patients suspected of PjP according to the final diagnosis retained. BDG values (**a**) with and (**b**) without evident cases of BDG false positive results. Comparison of the four groups was done using Kruskal-Wallis tests followed by Dunn’s pair-wise comparisons. *p*-adjusted values are indicated. Red line: median BDG value. **** *p* < 10^−4^.

**Table 1 jof-08-00663-t001:** Performances of BDG assay and molecular diagnosis for the diagnosis of PjP in 69 episodes from 67 HIV-positive patients as a screening strategy.

qPCR	Cut-Off (cp/mL)	Sensitivity (95% CI)	Specificity (95% CI)	PPV (95% CI)	NPV (95% CI)	LR
	3000	92.3% (75.9–98.6%)	93% (81.4–07.6%)	88.9% (71.9–96.1%)	95.2% (84.2–99.1%)	13.2
	30,000	92.3% (75.9–98.6%)	97.7% (87.97–99.9%)	96% (80.5–99.8%)	95.5% (84.9–99.2%)	39.7
**BDG**	**Cut-Off (pg/mL)**	**Sensitivity** **(95% CI)**	**Specificity (95% CI)/Specificity * (95% CI)**	**PPV (95% CI)/PPV * (95% CI)**	**NPV (95% CI)**	**LR/LR ***
	80	96.1% (81.1–99.8%)	81.4% (67.4–90.3%) 92.1% (79.2–97.3%)	75.8% (59–87.2%) 89.3% (72.8–96.3%)	97.2% (85.8–99.9%)	5.2/12.2
	200	88.5% (71–96%)	88.4% (75.5–94.9%) 97.4% (86.8–99.9%)	82.1% (64.4–92.1%) 95.8% (79.8–99.8%)	92.7% (80.6–97.5%)	7.6/34.5
	400	84.6% (66.5–93.8%)	95.3% (84.5–99.2%) 100% (91.4–100%)	91.7% (74.1–98.5%) 100% (85.1–100%)	91.1% (79.3–96.5%)	18.2/–

CI: confidence interval, PPV: positive predictive value, NPV: negative predictive value, LR: likelihood ratio, BDG: BD glucan, BPO: bronchopulmonary. * Excluding evident causes of BDG false-positive.

**Table 2 jof-08-00663-t002:** Performances of BDG assay and molecular diagnosis as a screening strategy for the diagnosis of PjP in 609 episodes occurring in 575 HIV-negative patients.

qPCR	Cut-Off (cp/mL)	Sensitivity (95% CI)	Specificity (95% CI)	PPV (95% CI)	NPV (95% CI)	LR
	3000	86.1 (71.3–93.9%)	97.4% (95.7–98.4%)	67.4% (53–79.1%)	99.1% (97.9–99.6%)	32.9
	30,000	63.9% (47.6–77.5%)	99.7% (98.7–99.9%)	92% (75–98.6%)	97.8% (96.2–98.7%)	183
**BDG**	**Cut-Off (pg/mL)**	**Sensitivity** **(95% CI)**	**Specificity (95% CI)/Specificity * (95% CI)**	**PPV (95% CI)/PPV * (95% CI)**	**NPV (95% CI)**	**LR/LR ***
	80	88.9% (74.7–95.6%)	82.6% (79.2–85.4%) 96.3% (94.3–97.7%)	24.2% (17.7–32.2%) 64% (50.1–75.9%)	99.2% (97.9–99.7%)	5.1/24.2
	200	72.2% (56–84.1%)	91.3% (79.7–96.6%) 99.2% (98.1–99.7%)	34.2% (24.5–45.4%) 86.7% (70.3–94.7%)	98.12% (96.6–99.0%)	8.3/95.1
	400	63.9% (47.6–77.5%)	94.4% (92.2–96%) 99.8% (99–100%)	41.8% (29.7–55%) 95.8% (79.8–99.8%)	97.6% (96–98.6%)	11.4/346.3

CI: confidence interval, PPV: positive predictive value, NPV: negative predictive value, LR: likelihood ratio, BDG: BD glucan, BPO: bronchopulmonary. * excluding evident causes of BDG false positive.

**Table 3 jof-08-00663-t003:** Performance of BDG assays for PjP and Pj colonization discrimination in 56 episodes occurring in 56 HIV-negative patients with a Pj low fungal load (<30,000 cp/mL).

Cut-Off (pg/mL)	Sensitivity (95% CI)	Specificity (95% CI)/Specificity * (95% CI)	PPV (95% CI)/PPV * (95% CI)	NPV (95% CI)	LR/LR *
80	83.3% (55.2–97%)	77.3% (63–87.2%) 89.5% (75.9–95.8%)	50% (29.9–70.1%) 71.4% (45.3–88.3%)	94.4% (81.9–99%)	3.7/7.9
200	58.3% (31.9–80.7%)	80.9% (78.8–96.4%) 100% (91.2–100%)	63.6% (35.4–84.8%) 100% (64.6–100%)	88.9% (76.5–95.2%	6.4/-
400	41.7% (19.3–68%)	95.4% (84.9–99.2%) 100% (91.6–100%)	71.4% (35.9–94.9%) 100% (56.5–100%)	85.7% (73.3–92.9%)	9.2/-

CI: confidence interval, PPV: positive predictive value, NPV: negative predictive value, LR: likelihood ratio, BDG: BD glucan. * excluding evident causes of BDG false positive.

## Data Availability

Detailed data are available on request within the limits relating to anonymization restrictions.

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
