# Peer review of "Usefulness of ß-d-Glucan Assay for the First-Line Diagnosis of Pneumocystis Pneumonia and for Discriminating between Pneumocystis Colonization and Pneumocystis Pneumonia"

_jof, 2022, doi:10.3390/jof8070663_

Round 1

Reviewer 1 Report

Usefulness of ß-D-Glucan assay for the first-line diagnosis of Pneumocystis pneumonia and for discriminating between first-line diagnosis colonization and Pneumocystis pneumonia

Jeanne Bigot, Sandra Vellaissamy, Yaye Senghor, Christophe Hennequin and Juliette Guitard

The authors review the utility of the ß-D-Glucan (BDG) test for the first-line diagnosis of Pneumocystis pneumonia (PjP) in immunocompromised patients, and they analyze the results according to whether these patients are HIV-infected or not. In the title, they also indicate the ability of this test to differentiate between colonization and invasive disease by this fungus.

They conclude that the determination of BDG in serum has a high sensitivity for the diagnosis of PjP in patient samples collected +/- 5 days around a positive qPCR test in bronchopulmonary samples.

INTRODUCTION

They introduce the difficulty in diagnosing PjP in immunosuppressed patients. Symptoms and imaging tests that raise suspicion of PJP have been described but are not conclusive, and in non-HIV patients they present more irregularly. The detection of fungal DNA by qPCR in bronchopulmonary samples is very sensitive, but difficult to interpret, and does not clearly differentiate colonization from invasion.

In this study, they discuss the utility of serum BDG for the initial diagnosis of PjP, while awaiting a qPCR result in bronchopulmonary samples, as well as a complementary support technique to differentiate colonization from invasion in patients with a low fungal load.

MATERIAL AND METHODS

Retrospective study of immunosuppressed patients (678 episodes according to the data in Figure 1) in whom qPCR for Pj DNA in bronchopulmonary samples and BDG (Fungitell, Cape Code) determination +/- 5 days around the bronchopulmonary sample had been performed. Patients are classified into 4 groups: Proven PjP, Probable PjP, Pj Colonisation and No Pj Infection.

COMMENTS OF MAJOR CONCERN about Patients/Episodes selection

The description of selected patients lacks an indication of the number of patients selected, to which group they belong (HIV vs. non-HIV), and the number of episodes assigned to each group.

In addition, pediatric services are mentioned in the acknowledgement section, so it would be convenient to include more details of the patients/episodes selected (perhaps as supplementary material). In the discussion, the results are only confronted with those published for adult patients, and we do not know whether pediatric patients, and their corresponding episodes, have been included in this study.

RESULTS

L. 117-119. This paragraph appears to be editorial recommendations to the authors. It should be removed.

L. 122. Figure 1 does not reflect the data described in the text (L. 121-122).

L. 127-128. The caption of Figure 1 does not describe its content, since the numerical data refer to episodes and not to patients. According to Figure 1, 678 episodes with qPCR and BDG data are analyzed, of which 69 correspond to HIV patients, and 609 episodes to non-HIV patients.

It would be helpful to show the flow chart, on the one hand, and on the other hand, the numerical data of patients per group, as well as the episodes assigned to each subgroup of patients.

L. 129-185. The description of the results of the HIV group refers to the number of episodes in the first lines (130-132), but from line 133 onwards results are mentioned in terms of patients, when they are episodes. We do not know the total number of patients in the group.

L. 147-148. The description in Table 1 should mention that data are derived from 69 "EPISODES" corresponding to "XX" patients in the HIV-positive group.

L. 165-168. At the bottom of Figure 2, the same indication as for Table 1.

L.177-185. At the end of this paragraph, reference is made for the first time to a number of patients (n=7), corresponding to samples with qPCR <30,000 cp/ml. It should be confirmed whether these are different patients, or different episodes.

L. 187-206 and 212-226. Same indication as for L. 129-185.

L. 207-208. In the description of Table 2, same recommendation as for Table 1.

L. 246-249. In the caption of Figure 3, same indication as for Table 1.

L. 251-265. Same indication as for L. 129-185.

L. 267-268. In the description of Table 3, same recommendation as for Table 1.

DISCUSSION and CONCLUSIONS

L. 272-275. This paragraph appears to be editorial instructions for the authors. I believe it should be deleted.

L. 300, 309, 321, 324-325. Verify that they refer to patients and not episodes.

The results are consistent with previous studies, and support the determination of serum BDG as a first-line test for the diagnosis of PjP, with a similar performance to qPCR of bronchopulmonary samples, and anticipating the results of the latter. However, they point out that it is necessary to verify the absence of confounding factors in patients with positive BDG (other concomitant fungal infections, surgical procedures, intravenous administration of immunoglobulins...).

In the same regard, they also conclude that serum BDG results can complement qPCR results, especially when qPCR samples have a low copy number, and thus differentiate between colonization and invasion. In the latter case, possible causes of false-positive BDG results should also be reviewed.

In addition to the limitations included in the manuscript, once the patient and episode data have been clarified, it should be mentioned that the diagnostic efficacy calculations could present some deviation due to the inclusion of two or more episodes of some patients, even though they are 30 days apart.

Finally, if episodes in pediatric patients have been included, this fact should also be taken into consideration when interpreting the results.

SUPPLEMENTARY MATERIAL

L. 355. In the description of Figure S1 the authors refer to patients, when they are "patient episodes".

L. 357. The figure only shows the comparison of 3 groups, not "four".

REFERENCES

The list is up to date and there is only one self-citation (ref.6), necessary for the comparison of techniques.

ENGLISH LANGUAGE

It is appropriate, only minimal errors are detected.

L. 152: "...with suspected of PjP..." delete "of".

L. 263: "...with an evident of false positive result..." delete "of".

L. 347: "...may be use to anticipate...", change to "used".

L. 358: correct the spelling for "multiple".

Reviewer 2 Report

II  In this manuscript, the authors conducted a retrospective study to investigate the performance of serum β-D-glucan (BDG) assay for the diagnosis of Pneumocystis jirovecii pneumonia (PJP) in immunocompromised patients, with and without HIV infections. They also evaluated the potential value of BDG test to differentiate PJP infection from PJP colonization among patients with uncertain diagnosis by qPCR method. This study is well written and has a great interest for clinical practice. The study design and methods are overall appropriate. However, there are some concerns that I hope the authors can address: 

T  The study patients were all immunocompromised patients (Line 66) and after sample screening, 69 episodes from HIV+ patients and 609 episodes from HIV- patients were included in the data analysis. I would like to know the following information: How many patients did all these episodes (69+609) come from?  What kind of immunocompromised patients were they?  Could the authors provide basic data about the underlying conditions of these patients? Such information would provide a better understanding of this study.   

q  qPCR and BDG are both quantitative methods to diagnose PjP. It would be interesting to know if there is a correlation between qPCR and BDG values in the proven/probable PjP episodes. Also, it would be interesting to know if HIV+ patients and HIV- patients made a difference in such correlation in their proven/probable PjP episodes. Have the authors investigated this question?

F   Figure 2 is about comparing BDG values in HIV+ patients among groups with different PjP diagnosis results. The authors stated that the comparisons were done by ANOVA test followed by Kruskal-Wallis test (Line 167-168). If true, this is problematic in statistics. Kruskal-Wallis test is actually the nonparametric equivalent of ANOVA test. If BDG data is normally distributed, then ANOVA test should be used here to compare means among the groups. On the other hand, if BDG data is not normally distributed, then Kruskal-Wallis test should be used to compare medians. The two tests are not supposed to be used for the same data comparison. In general, following either ANOVA or Kruskal-Wallis test (depending on data distribution) for multiple comparisons, pairwise comparisons should be performed using an appropriate test with an alpha-adjustment.  Similar problem for Figure 3.  

F For Figures 2 and 3, I would suggest indicating the median BDG value of each group on the figures.  For instance, the authors can highlight them in a bright color.

L  Line 173-174: In HIV+ patients, when increasing the BDG cut-off to 200 pg/ml and 400 pg/ml, the authors stated that the specificity was increased to 97.4% [95% CI: 86.8-99.9%] and 100% [95% CI: 91.4-100%], respectively. These values, however, are the calculated specificities after excluding the episodes that may have led to false positive results. Please indicate this exclusion in the description.

L  Line 304: In discussion, the authors stated that the study results are overall in accordance with a meta-analysis, in which the calculated sensitivity and specificity of BDG were ”88.9% and 82.6, respectively” for HIV- patients. These values are the sensitivity and specificity of BDG test in HIV- patients reported in the authors’ study. In the meta-analysis the authors mentioned, the sensitivity and specificity of BDG in HIV- patients were 86% and 83%, respectively.

L Line 327: The authors stated that for HIV- patients with low fungal burden (n=56), the sensitivity increased from 83.3% to 89.5% after excluding the 6 episodes with evident cause for false-positive BDG results. This sensitivity change seems to be impossible to me. According to the sensitivity definition, sensitivity of BDG test = number of episodes with positive BDG results / number of proven or probable PjP episodes. As we can see, the sensitivity calculation is done inside the group of proven/probable PjP episodes (the denominator of the calculation). Although the 6 episodes with evident cause for false positive BDG results had high BDG values, they were not proven/probable PjP episodes and therefore they are outside the group of episodes that are used for the sensitivity calculation. Then how could the episodes with false positive BDG results affect the BDG sensitivity?

L  Line 117-119 and Line 272-275  Those sentences should be removed. They seem to be the guidance for writing certain sections and are not relevant to the study.

Reviewer 3 Report

The paper by Bigot and et al are well written. The authors evaluated the performances of the BDG assay in two clinical indications: (i) as a first-line diagnostic test in immunocompromised patients suspected of PjP before Pj qPCR result is available (ii) as an aid to distinguishing between Pj colonization and infection in patients presenting with a low fungal load detected with qPCR applied to a bronchopulmonary sample.

The article is interesting, however there are some points that should be improved:

Line 11: To change PjP to PcP.

Line 45: The authors show a single reference and they say that there are several studies, please add others studies.

Line 58: The comparison of PCR vs. BDG studies should be mentioned. In the same way, the authors should mention the studies carried out with BDG in HIV-negative patients.

Line 67: How many samples (BALF and IS) are in each patient and each group?.

Line 68: To add the place where the study was developed.

Line 83: To add one reference.

Line 84:  How is possible that for different samples the cut-off value is the same? Please clarify this idea.

Line 137: Why the authors explain of target mutation? Another gene was evaluated?, it cannot be an inhibition of the sample. Please clarify this aspect.

Line 138: The authors must explain the discordance of the results, please clarify this idea in discussion section.

Line 145: The authors must incorporate the calculation of the Jouden Index (IJ).

Line 145: The authors must incorporate the Kappa index (qPCR vs BDG).

Line 276: The authors show a single reference and they say that there are several studies, please add others studies.

The authors have to update the references.

Round 2

Reviewer 1 Report

The authors have made a great effort to adapt to the reviewers’ comments, but there remain some issues that could improve the final version.

FIGURE 1 has been reorganized, but I would suggest the following:

1. Introduce the lines describing patients, episodes and specimens in this order

            - HIV-positive patients (n)                                   17  

            - Episodes (n)                                                     17   

            - qPCR specimen type analyzed (IS/BALF)      3 / 14  

And the same for Immunocompromised HIV-negative patients

2. The inclusion of pediatric patients should be mentioned. You could add a superindex next to the figures of the two groups of non-VIH patients that include pediatric samples, and explain this issue in the footnote            

Probable PjP patients     28a   (in the footnote: a including one pediatric patient)

            Non-PjP                       493 b  (in the footnote: b including 28 pediatric patients)

3. The legend of Figure 1 does not explain its content. Maybe you could say something like “Characteristics and classification of patients/episodes included in the study”.

TABLES

In general, all tables require adaptin their format to fit columns and lines properly.

ENGLISH LANGUAGE

The new paragraphs require a revision of the English language, and some of them are difficult to interpret (highlighted in green).

And some words appear misspelled (some of them are highlighted in green in the revised version of the manuscript). 

I'm sending a marked version of the revised manuscript
